# Pequi Oil (*Caryocar brasiliense* Camb.) Attenuates the Adverse Effects of Cyclical Heat Stress and Modulates the Oxidative Stress-Related Genes in Broiler Chickens

**DOI:** 10.3390/ani13121896

**Published:** 2023-06-06

**Authors:** Jéssica Moraes Cruvinel, Priscila Michelin Groff Urayama, Cássio Yutto Oura, Fernanda Kaiser de Lima Krenchinski, Tatiane Souza dos Santos, Beatriz Alves de Souza, Samir Moura Kadri, Camila Renata Correa, José Roberto Sartori, Antonio Celso Pezzato

**Affiliations:** 1Department of Breeding and Animal Nutrition, School of Veterinary Medicine and Animal Science, São Paulo State University (UNESP), Doutor Walter Mauricio Correa s/n, Botucatu 18618-681, SP, Brazil; priscilagroff@hotmail.com (P.M.G.U.); cassioyoura@gmail.com (C.Y.O.); balanca.armazemrondon@gmail.com (F.K.d.L.K.); tatianesouza.santos@yahoo.com.br (T.S.d.S.); bia.as.souza@gmail.com (B.A.d.S.); jose.sartori@unesp.br (J.R.S.); celso.pezzato@unesp.br (A.C.P.); 2Department of Animal Production and Preventive Veterinary Medicine, School of Veterinary Medicine and Animal Science, São Paulo State University (UNESP), Botucatu 18610-034, SP, Brazil; samir.kadri@unesp.br; 3Department of Pathology and Experimental Research Unit (UNIPEX), Medical School, São Paulo State University (UNESP), Distrito Rubião Jr., Botucatu 18618-970, SP, Brazil

**Keywords:** antioxidant, broiler, carotenoids, oxidative damage, pequi

## Abstract

**Simple Summary:**

Pequi (*Caryocar brasiliense* Camb.) is an evergreen tree typical of the biodiversity in the Brazilian cerrado biome and it represents an important source of income for communities that sell its fruit and related products, such as PO (pequi oil). Pequi oil has antioxidant properties as a result of its high concentration of carotenoids, and said properties have been investigated in in vitro studies and animal models. However, up until now, the effects of dietary supplementation with PO have not been investigated in broiler chickens, which raised our research interest. We found that the levels of pequi oil used in broiler chickens submitted to high temperatures had a hepatoprotective effect; in addition, higher levels reduced the concentration of malondialdehyde in their livers. Corroborating these results, birds fed with higher levels of pequi oil showed a 92% reduction in the concentration of their Hsp 70 mRNA in comparison to birds with supplementation, whereas the Nrf2 gene was upregulated (37%). Therefore, supplementation with PO relieves alterations in the antioxidant system caused by heat stress in broiler chickens, acting as a potential antioxidant additive for use in poultry production. Furthermore, our findings can inform future studies.

**Abstract:**

The present study was conducted to determine the possible antioxidant protection of pequi oil (PO) against cyclic heat stress in broiler chickens and to highlight the application of PO as a promising additive in broiler feed. A total of 400 one-day-old male broiler chicks (Cobb 500) were randomly assigned to 2 × 5 factorially arranged treatments: two temperature-controlled rooms (thermoneutral—TN or heat stress—HS for 8 h/day) and five dietary PO levels (0, 1.5, 3.0, 4.5, or 6.0 g/kg diet) for 42 days. Each treatment consisted of eight replicates of five birds. The results showed that HS increased glucose (*p* = 0.006), triglycerides (*p* < 0.001), and HDL (*p* = 0.042) at 21 days and reduced (*p* = 0.005) serum total cholesterol at 42 days. The results also showed that HS increased the contents of alanine aminotransferase (ALT) and aspartate aminotransferase (AST). In contrast, PO linearly decreased AST (*p* = 0.048) and ALT (*p* = 0.020) at 21 and 42 days, respectively. The heterophil-to-lymphocyte ratio in the birds under HS was higher than in those in the TN environment (*p* = 0.046). Heat stress decreased (*p* = 0.032) the relative weight of their livers at 21 days. The superoxide dismutase activity increased (*p* = 0.010) in the HS treatments in comparison to the TN treatments, while the glutathione peroxidase activity in the liver decreased (*p* < 0.001) at 42 days; however, the activity of catalase had no significant effects. Meanwhile, increasing the dietary PO levels linearly decreased plasma malondialdehyde (*p* < 0.001) in the birds in the HS environment. In addition, PO reduced (*p* = 0.027) the expression of Hsp 70 in the liver by 92% when compared to the TN treatment without PO, mainly at the 6.0 g/kg diet level. The expression of Nrf2 was upregulated by 37% (*p* = 0.049) in response to PO with the 6.0 g/kg diet compared to the HS treatment without PO. In conclusion, PO supplementation alleviated the adverse effects of HS on broilers due to its antioxidant action and modulation of the genes related to oxidative stress, providing insights into its application as a potential feed additive in broiler production.

## 1. Introduction

Heat stress (HS) in poultry farming is a worldwide issue, particularly in tropical and subtropical areas [1,2]. In addition, the high efficiency of meat production due to genetic advancements means that broiler chickens have a rapid metabolic rate and, consequently, a high heat production, which makes them susceptible to HS [3]. Under such conditions, the balance between the generation and elimination of reactive oxygen species (ROS) may be compromised, leading to a depletion in antioxidant defenses and, in turn, an induction of cellular oxidative stress [4,5].

Animal cells have developed a sophisticated defense system that is able to neutralize ROS [6,7], featuring antioxidant enzymes (e.g., catalase, dismutase superoxide, and glutathione peroxidase), non-enzymatic antioxidants (e.g., vitamins, minerals, carotenoids, and glutathione), and a modulation of the expression of cytoprotective genes, such as thermal shock proteins (Hsp 70) and the nuclear factor erythroid 2-related factor 2 (Nrf2). However, such a system is effective up to a certain stage, especially in face of chronic exposure to heat, which is often the case in poultry farming and leads to large economical losses, as it depletes the antioxidant defenses of the animals and, consequently, compromises the animal productivity [8]. In these situations, a greater supply of antioxidants to recover the equilibrium of the redox system and prevent oxidative damage becomes essential in the productive system. This makes the use of natural products in broiler nutrition a possible strategy for mitigating HS, besides regulating the expression of important genes in the redox system and increasing the cellular defense capability [5,7,9].

Pequi (*Caryocar brasiliense* Camb.) is an oily fruit native to the Brazilian cerrado biome and is considered to be one of the species of the greatest socioeconomic relevance to the region [10]. Pequi trees tolerate long periods of direct insolation, are resistant to moisture, drought, and heat, and can adapt to a variety of soils [11]. Their fruits (pequi) contain one or more seeds covered by fleshy and aromatic pulp of a slightly sweet taste [12]. The oil extracted from the pulp (yellow/orange color) is rich in unsaturated fatty acids (e.g., oleic acid; ω9) and carotenoids such as β-carotene, lycopene, and lutein [11,12,13]. The antioxidant, anti-inflammatory, cytoprotective, and antitumoral properties of pequi oil (PO) have been reported in animal models and/or in vitro studies [10,14]. Among those, the antioxidant properties are particularly appealing, since PO has a high carotenoid content.

Carotenoids are natural pigment molecules that are also able to inactivate the electronically excited molecules involved in the generation of radicals (singlet oxygen) and the regeneration of biomolecules damaged by oxidative lesions [15]. Miranda-Vilela et al. [14] showed that PO was able to reduce the effects of oxidative stress induced by chemotherapy drugs in the liver cells of mice with Ehrlich tumors, an effect even greater than experimental models of vitamins C and E. Likewise, Vale et al. [16] reported that the oral administration of PO to Wistar rats alleviated the oxidative stress induced by exhaustive physical exercise, maintaining their levels of non-enzymatic antioxidants and inhibiting their lipid peroxidation, characterized by lower a concentration of malondialdehyde (MDA). However, to date, the effects of dietary supplementation with PO have not been investigated in broiler chickens or other domestic animal species.

Therefore, the current study was conducted to investigate the effects of graded levels of PO supplementation on the plasma biochemical indices, relative organ weights, antioxidant statuses, and gene expressions of Hsp 70 and Nrf2 in broiler chickens subjected to cyclic heat stress. The results of this research could provide guidance and a reference for the further application of PO as a promising phytogenic feed additive in broiler feed.

## 2. Materials and Methods

All the experimental procedures were approved by the Animal Ethics Committee of the Veterinary and Animal Science College, São Paulo State University, Botucatu, SP, Brazil (protocol 0192/2018—CEUA).

### 2.1. Pequi Oil (Caryocar brasiliense Camb.)

*C. brasiliense* fruits were collected during the harvesting period in January 2019. The oil was extracted from the pulp using a manual process employing hot water and was purchased from Grande Sertão Cooperative (Montes Claros, MG, Brazil). The pequi oil composition is shown in Table 1.

### 2.2. Chicks, Housing, and Management

A total of 400 one-day-old commercial male broiler chicks (Cobb 500) with an average initial body weight of 47.5 ± 1.27 g were obtained from a local commercial hatchery (Pluma, Descalvado, SP, Brazil). The broilers were vaccinated upon hatching for Marek’s disease, infectious bronchitis, and Gumboro disease. On day 1, the broiler chicks were assigned in a fully randomized design with a 2 × 5 factorial arrangement to treatments with two temperature-controlled rooms (thermoneutral room—TN; heat stress room—HS) and five dietary pequi oil (PO) levels, with eight replicates of five birds each. The experimental period lasted for 42 days.

In each of the temperature-controlled rooms, the birds were housed in 40 battery cages (8 cages/treatment) with wire floors (0.4 × 0.5 × 0.6 m, height × length × width). Water was supplied in nipple drinkers and trough feeders made of galvanized steel were placed in front of the cages. The experimental diets (in mash form) and fresh water were provided ad libitum. The lighting program was applied according to the management guides and recommendations of the company [19].

### 2.3. Experimental Diets and Heat Challenge

The experimental diets were prepared by including levels of PO into the basal diets. The five dietary treatments were as follows: control diet (CON; basal diet without PO), CON + 1.5 g PO/kg diet; CON + 3.0 g PO/kg diet; CON + 4.5 g PO/kg diet; and CON + 6.0 g PO/kg diet. The PO was first mixed intensively with the associated soybean oil and then gradually added to the diet. The basal diet was formulated to correspond to nutrient requirements that were equal to or slightly lower than those recommended by Rostagno et al. [19] for broilers, and the feeding program consisted of pre-starter (days 1 to 7), starter (days 8 to 21), grower (days 22 to 35), and finisher (days 36 to 42) (Table 2 and Table 3). The formulation of the diets considered the apparent metabolizable energy (AME) of PO (7370 kcal/kg) based on data from a previous study. 

The birds were reared in temperature-controlled rooms with independent temperature control. During the experimental period (from 1 to 42 days of age), all the broiler chickens in the thermoneutral room (TN) were reared under the temperature conditions recommended by the Cobb Broiler Management Guide [19], but adapted to the Brazilian context. The broilers from the remaining groups were kept in another room and exposed daily to 8 h (8:00 a.m. to 4:00 p.m.) of cyclical heat stress (HS), after which, the temperature was lowered to the same level as that of the TN group for the remaining 16 h over the 42 days of the experiment. Each room was equipped with portable, automatically controlled electric heaters (1500 W) and was provided with a fan for the circulation of this hot air. The temperature and relative humidity were recorded daily and the average temperature and relative humidity were calculated. The temperature scheme is shown in Table 4.

### 2.4. Blood Parameters and Determination of Serum MDA

At 21 and 42 days of age, blood samples were collected from the broiler chicks (n = 8 birds per treatment) into clean sterile tubes and left to coagulate before being centrifuged at 3500 rpm for 15 min to separate the serum. The serum samples were stored in Eppendorf tubes at −20 °C until the analysis. The following parameters were determined spectrophotometrically in the serum using commercial reagent kits (LaborLab, Guarulhos, SP, Brazil) via a semi-automated biochemical analyzer (BIO-200S, *Bioplus Produtos para Laboratórios* Ltd.a, Barueri, SP, Brazil): glucose (mg/dL; no. K133), total cholesterol—CHO (mg/dL; no. K083), triglycerides—TG (mg/dL; no. K117), high-density lipoprotein—HDL (mg/dL; no. K071), aspartate aminotransferase—AST (U/L; no. K048), and alanine aminotransferase—ALT (U/L; no. K049).

On day 42, one bird per cage (n = eight birds per treatment) was sampled at random. Blood samples were taken with a 23-gauge needle from the jugular vein and collected into two tubes. The first tube, containing EDTA, was centrifuged (3500 rpm, 15 min) to obtain plasma for a determination of the leukocyte profile. The second tube, containing heparin, was used to obtain the serum malondialdehyde (MDA) concentrations and was centrifuged at 3500 rpm for 15 min at 4 °C before being stored at −80 °C until further analysis. For the leukocyte population, the blood samples were stained according to Lucas and Jamroz [20,21], and subsequently, 100 leukocytes per sample were counted using an optical microscope (BX51; Olympus, Tokyo, Japan). The heterophil-to-lymphocyte ratio (H/L) for each bird was also calculated. The serum MDA concentrations were determined using the TBA (thiobarbituric acid) method with absorbance at 532 nm, according to Buege and Aust [22]. The results were expressed as nanomoles per liter (nmol/L) and each sample was analyzed in duplicate. 

### 2.5. Organ Weights

At 21 and 42 days, one bird per cage (n = eight per treatment) was euthanized via cervical dislocation and the whole spleen, bursa of the Fabricius, thymus, pancreas, and liver were immediately removed and weighed. The relative weights of these organs were expressed as percentages of the live weight of the birds.

### 2.6. Measuring Activities of Antioxidant Enzymes in the Liver

On day 42, the broilers were slaughtered via cervical dislocation and liver samples were collected (n = 8 samples per treatment), immediately immersed in liquid nitrogen, and stored at −80 °C until the activity of their antioxidant enzymes was determined (SOD, CAT, and GSH-Px). For the activities of SOD (superoxide dismutase; EC 1.15.1.1), CAT (catalase; EC 1.11.1.6), and GSH-Px (glutathione peroxidase; EC 1.11.1.9), 1 g of a liver sample was homogenized in 5 mL of 0.05 M phosphate buffer (pH = 7.0). The homogenate was centrifuged at 5000 rpm for 20 min at 4 °C. The supernatant was used for the enzyme activity and protein content analyses. The activities of SOD, CAT, and GSH-Px were determined as previously described by Vicente et al. [23]. The absorbance was read on a spectrophotometer at 560 nm, 610 nm, and 420 nm, respectively. For all the antioxidant enzyme assays, the total protein concentration was determined using the method of Bradford [24] with Coomassie Brilliant Blue G-250 dye, using bovine serum albumin as a standard.

### 2.7. RNA Extraction and Quantitative Real-Time PCR Analysis

RNA was extracted from a 50 mg liver sample from four birds per treatment [25]. The extraction was performed using the Trizol method with 500 µL of TRIzol (Invitrogen, Carlsbad, CA, USA) for each sample, in order to disrupt the cells and release their contents. The extraction product was visualized on 1% agarose gel and quantified using a NanoDrop 1000 spectrophotometer (ThermoFisher Scientific, Waltham, MA, USA). Next, all the samples were stored at −80 °C until they were ready to use. The samples were then treated with DNase, and the cDNA synthesis reaction was set as follows: a mix of 0.75 mM of oligo(dT) solution (n = 18); 0.15 mM of random oligonucleotides (n = 8); 0.75 mM of dNTP, and 11 µL of RNA, which was treated with DNAse in the previous step and prepared and incubated at 65 °C for 5 min, then placed on ice for 1 min. For this preparation, 0.5 mM of DTT, 40 U of RNaseOUT, and 100 U of SuperScript III were added. The reaction was then incubated at 50 °C for 1 h and then at 70 °C for 15 min. 

After this, a real-time PCR analysis (RT-PCR) was performed using an Applied Biosystems StepOnePlus Real-Time PCR System (Applied Biosystems, Foster City, CA, USA) and a SYBR Green PCR Master Mix kit (Applied Biosystems, Foster City, CA, USA). The primer sequences for the target and reference genes are listed in Table 5. The PCR cycle parameters were as follows: one cycle at 50 °C for 2 min, one cycle at 94 °C for 10 min, and 40 cycles of 94 °C for 15 s and 60 °C for 1 min. The dissociation curve was obtained as follows: 95 °C for 15 s, 60 °C for 30 s, and 95 °C for 15 s. To calculate the efficiency of the oligonucleotides used, four dilutions of cDNA samples were performed at 1:5, 1:25, 1:125, and 1:625. The efficiency (E) was calculated using the formula E = 10 (−1/slope). The relative levels of the mRNA expression were calculated using the 2^−ΔΔCT^ method [26], which was normalized to the reference mRNA level of β-actin. Each sample was analyzed in triplicate.

### 2.8. Statistical Analysis

The data were analyzed as a completely randomized design using the MIXED procedure in the software SAS (version 9.2, SAS Institute Inc., Cary, NC, USA) [27]. The model used was y_ij_ = μ + T_i_ + PO_j_ + (T × PO)_ij_ + e_ij_, where y = the response variable, µ = the population mean, Ti = the main effect of the temperature, PO = the main effect of the dietary pequi oil level, T × PO = the interaction effect of the temperature with the dietary pequi oil levels, and e_ij_ = the residual error. Orthogonal polynomial contrasts were also applied to determine the linear and quadratic responses to the different levels of PO supplementation. The results of the relative expressions of each target gene in the HS treatments, i.e., Hsp 70 and Nrf2, were compared to the treatment with no PO under thermoneutral ambient conditions (TN; control). The significant differences among the treatment means were compared using Tukey’s test. The differences were considered statistically significant when the *p*-value was less than 0.05. The results were presented as means with their pooled standard errors.

## 3. Results

### 3.1. Blood Parameters

No effect of the interaction of the dietary PO levels with the environmental temperature was found on the serum biochemical parameters (Table 6). On day 21, heat stress (HS) significantly increased the serum glucose (*p* = 0.006), TG (*p* < 0.001), and HDL (*p* = 0.042) concentrations in comparison to those of the birds housed in the thermoneutral (TN) condition. Additionally, variations in the serum CHO in response to the dietary PO levels were independent of the environmental temperature (*p* = 0.003). On day 42, the HS condition significantly reduced (*p* = 0.005) the serum CHO and did not change the plasma glucose, TG, or HDL concentrations. Furthermore, the birds challenged by HS had an increased serum enzyme activity of ALT at day 21 (*p* = 0.020) and increased activities of AST (*p* = 0.012) and ALT (*p* = 0.015) at day 42 in comparison to the TN condition. Supplementation with PO linearly reduced the serum enzyme activities of AST (linear: *p* = 0.048; y = 236.77−6.90x; R^2^ = 0.59) and ALT (linear: *p* = 0.020; y = 5.343−0.407x; R^2^ = 0.87) at days 21 and 42, respectively.

The data on the leukocyte subsets are shown in Table 7. On day 42, no differences were observed in the blood leukocyte profiles among all the experimental treatments, except for the basophil counts in response to the dietary PO levels (*p* = 0.022). However, increases in the H/L in response to the environmental temperature were independent of the dietary PO levels (*p* = 0.046). 

### 3.2. Relative Organ Weights

Table 8 summarizes the relative organ weights of the birds subjected to different environmental temperatures and diets containing different PO levels. Compared to the TN condition, exposure to HS had no significant effect on the relative weights of the spleen, bursa of the Fabricius, thymus, or pancreas, but decreased (*p* = 0.032) the relative weight of the liver at day 21. The dietary PO levels had no significant effect on the relative organ weights. 

### 3.3. Antioxidant Activities of Liver Enzymes and Serum Lipid Peroxidation

No effect of the interaction between the environmental temperature and dietary PO levels on the antioxidant activity of the enzymes in the broilers was found (Table 9). The SOD activity increased (*p* = 0.010) in response to exposure to HS, whereas the GSH-Px activity decreased (*p* < 0.001) when compared to the birds housed under the TN condition. No significant difference was observed (*p* > 0.05) in the hepatic CAT activity among all the treatments. However, the serum MDA concentration responses to the increasing dietary PO levels in the birds housed in either the TN or HS conditions were different (interaction effect: *p* = 0.012). The serum MDA content was 42.5% lower in the broilers that received supplementation with 6.0 PO under the HS condition than in those that were only challenged with HS (18.99 vs. 10.93 nmol/L). Furthermore, the reduction in the MDA concentration was more notable in the HS environment with broilers supplemented with PO than in the TN environment. Increasing dietary PO levels linearly decreased the MDA concentrations (linear HS: *p* < 0.001; y = 18.086 − 1.554x; R^2^ = 0.59) in the birds housed in the HS condition (Table 10).

### 3.4. Heat Shock Protein 70 and Nrf2 Expressions in the Liver

Figure 1 shows the relative mRNA expressions of Nrf2 and Hsp 70 in the livers of the birds. Compared to the TN treatment without PO, the expression levels of the Hsp 70 gene were significantly upregulated (*p* = 0.027) in response to the HS environment. The dietary PO levels reduced the liver Hsp 70 levels by 92% compared to the birds reared in the HS environment, mainly at the 6.0 g PO/kg diet level (4.02 vs. 0.32 relative gene expression). In addition, the hepatic Nrf2 levels were not affected in the HS groups compared to those in the TN treatments. However, compared to the HS treatment without PO, the treatment using the 6.0 g PO/kg diet enhanced the liver Nrf2 expression by 37% (*p* = 0.049; 2.06 vs. 1.30 relative gene expression).

## 4. Discussion

The present study was carried out to investigate the hypothesis that supplementation with PO alleviates alterations in the antioxidant system caused by heat stress (HS) in broiler chickens. The high cyclical temperatures during the day in the HS rooms were above the recommendations for the lineage according to the rearing phases [16]. In poultry, when the ambient environmental temperature exceeds the thermos neutral zone (16–25 °C), thermal injury is initiated [1,2,3,4,5,6,7]. In addition, the panting behavior and longer times of rest observed suggest that the broilers were submitted to HS conditions during this experiment. 

The results of this study show that the birds under HS had lower plasma concentrations of glucose, triglyceride, and HDL at 21 days of age, which corroborates the results observed by Puvadolpirod and Thaxton [28]. Animals under HS use glucose and gluconeogenic precursors (e.g., glycerol, glucogenic amino acids, and lactate), as well as the cleavage of reserve triglycerides, as their main sources of energy [29,30,31], which is a strategy for the organism to maintain their body temperature and re-establish homeostasis via an energy adjustment. However, such data were not impacted at 42 days of age. It can be reasoned that the stress initiated from one day of age may have induced thermotolerance in the face of transformations in the hypothalamic regions, as proposed by Yahav and McMurtry [32] and Kisliouk et al. [33]. 

The levels of ALT and AST in circulation may be high when liver damage occurs; thus, these enzymes are employed as sensitive markers of hepatocellular lesions [34]. The present study observed that HS led to increases in the serum levels of ALT and AST in the broiler chickens, which suggests that liver lesions were induced. However, PO had an hepatoprotective effect under HS conditions in the face of the linear reduction in the serum levels of ALT and AST. Consistent with such findings, Miranda-Vilella et al. [14] and Colombo et al. [35] reported that oral supplementation with PO in rats treated with chemotherapy drugs to induce oxidative stress reduced the liver lesions caused by high ROS concentrations.

The heterophil–lymphocyte ratio (H/L) is a primary indicator of stress in birds [1]. In agreement with several studies [36,37], it was observed that the broilers exposed to high temperatures exhibited an increased H/L; however, the PO levels did not impact the other blood parameters. Variations were observed in the amount of basophils in the PO levels. Nonetheless, such data were not correlated with any specific PO dose and the basophil values were within the range expected for the lineage studied, as described by Sturkie [38] and Bounous and Stedman [39].

Some studies have shown that HS decreases the relative weights of immunologic organs (the thymus, spleen, and bursa of the Fabricius in chickens) [1,40]. The present study found that HS only significantly decreased the relative weight of the liver at 21 days of stress, similar to the results obtained by Zhang et al. [7]. The relative weight of an organ may reflect its growth and development to some degree. In addition, some phytogenic additives act indirectly by inactivating free radicals, thus allowing for the mitigation of negative effects and, consequently, the maintenance of an adequate immune status under high temperature conditions [41]. Given the lack of alterations in the relative weights of the organs in the present study, further investigations are suggested under different conditions to assess the action of PO on cellular and humoral immunity. 

As the temperature increased, the activities of the antioxidant enzymes in the liver and plasma may have been upregulated as a protective response against oxidative stress [42]. Likewise, it was observed that HS increased the SOD activity in the liver. The greater SOD activity in the HS group was likely due to the high production of the superoxide anion, which is considered to be the primary product of the ROS production system [43]. The superoxide anion is easily dismutated into hydrogen peroxide in a reaction catalyzed by SOD. The breakdown of hydrogen peroxide (H_2_O_2_) into water is regulated by the CAT and GSH-Px enzymes, but the GSH-Px activity was reduced by HS in this study. This allows for the inference that H_2_O_2_ takes part in the formation of more reactive radicals, such as the hydroxyl radical in Fenton reactions [44]. It might be the case that HS causes protein denaturation, since GSH-Px is associated with protein protection via a mechanism known as protein S-glutationylation [45]. Such a sequence of reactions may partially explain the alterations in the concentrations of these antioxidant enzymes in the liver. 

The antioxidant action of PO was able to linearly reduce the lipid peroxidation in the broilers submitted to HS, which was a result of the decrease in the MDA content, particularly at the PO level of 6.0 g/kg diet (18.99 HS vs. 10.93 TN nmol/L). In accordance with this finding, Vale et al. [16] reported that diets supplemented with PO (400 mg/five weeks) in a group of Wistar rats under a strenuous exercise protocol significantly depressed this lipid peroxidation. In other studies, Sahin et al. [46] and Rajput et al. [47] showed that supplementations in chicken diets with lutein (200 mg/kg) and lycopene (200 or 400 mg/kg), respectively, were able to reduce the serum concentrations of MDA. MDA is a byproduct of lipid peroxidation, and its serum concentration can be used as an indicator of cellular peroxidation and ROS accumulation [3,48]. It is assumed the results obtained in the present research were associated with the antioxidant compounds (carotenoids) present in PO, shown in Table 1. This table shows the quantification of the carotenoids in PO, characterizing it as an excellent source of antioxidant compounds, mainly beta-carotene, lycopene, and lutein, which act in ROS neutralization [49,50]. These lipophilic molecules exert their protective effects associated with proteins+ and lipoprotein structures in cell membranes via mechanisms to reduce oxidants [51,52] and regenerate biomolecules in the prevention and interruption of the lipid peroxidation cascade [15].

Exposure to HS upregulates the synthesis of the HSPs (heat shock proteins) produced in all cells and tissues in response to stress, which are aimed at facilitating the production, conformation, and renovation of other proteins [53,54]. The expression of Hsp 70 is a classical sign of stress due to high temperatures in birds [55,56]. On the other hand, ROS overproduction is also involved in inducing Hsp 70 synthesis [9], which corroborates the present study. The pequi oil levels were able to act in the regulation of the Hsp 70 expression, since PO supplementation decreased the liver levels of this gene in birds under HS, particularly at the PO level of the 6.0 g/kg diet. Such a finding is another indication that PO acts as a cytoprotective agent in broiler chickens subjected to high temperatures.

To further explore the mechanisms subjacent to the antioxidant effects of PO, the Nrf2 expression in the liver was determined. Nrf2 is a sensitive redox transcription factor that regulates the numerous genes that code phase II detoxifying enzymes and antioxidant enzymes such as GSH-Px, SOD, CAT, heme oxygenase-1, and glutathione-S-transferase [7]. The present study found that the PO level of the 6.0 g/kg diet increased the Nrf2 expression (37%) in the broiler chickens under HS in comparison to those not supplemented with PO in HS conditions. Thus, the bioactive compounds in PO stimulated the transcription and translation of the cytoprotective proteins in liver cells. In line with these findings, Zhang et al. [7] and Sahin et al. [46] reported that the activation of the liver Nrf2 level increased in a dose-dependent manner in birds supplemented with lycopene (200 or 400 mg/kg) and turmeric (50, 100, or 200 mg/kg), respectively, under high-temperature conditions. The antioxidant properties of several natural properties are able to modulate the Nrf2 system, aiding in the maintenance of health and the overall cell oxidation–reduction equilibrium. However, the mechanisms of each compound involved are not well established [7]. Further research to explain these effects of PO on animal production is relevant. Nevertheless, the results of the present study suggest that PO inclusion, starting at a 6.0 g/kg diet, would be a promising dietary additive in broiler feed.

## 5. Conclusions

In conclusion, adding PO, especially at the level of a 6.0 g/kg diet, alleviated the oxidative stress induced by a high ambient temperature and had a hepatoprotective effect in broiler chickens. Moreover, the PO supplementation resulted in low levels of Hsp 70 expression and induced the expression of Nrf2-mediated genes. Overall, such results suggest supplementing broiler chickens with PO in a 6.0 g/kg diet may activate the defense mechanisms of the organism, thus decreasing the molecular oxidative effects. Such antioxidant actions of PO alleviated the negative effects of HS, particularly when the birds were exposed to stress in hot climates, acting as an excellent antioxidant additive in poultry production. 

## Figures and Tables

**Figure 1 animals-13-01896-f001:**
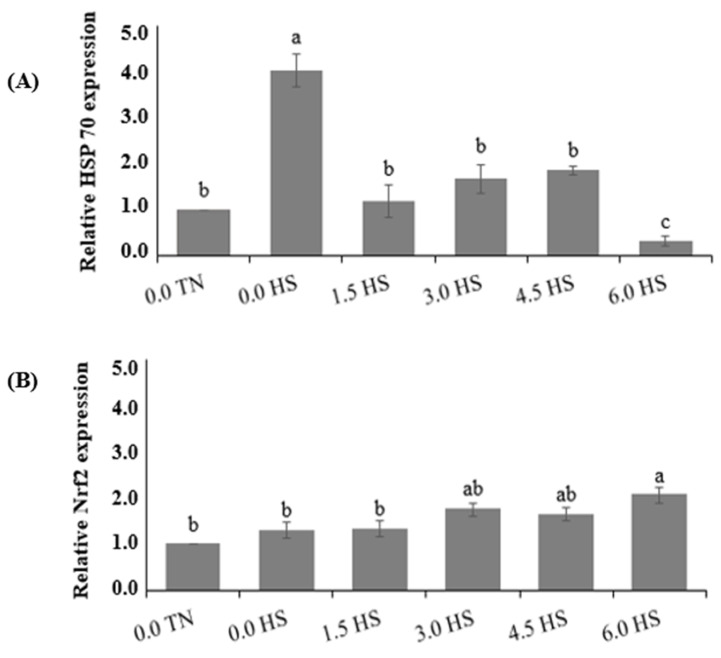
Effect of PO on the Hsp70 and Nrf2 expressions in broilers livers at 42 days of age. (**A**) Hsp70; and (**B**) Nrf2. Each bar represents the mean ± SEM (n = 4). Bars not sharing the same letters (a–c) are significantly different according to Tukey’s comparison test at *p* < 0.05. 0.0TN = normal temperature + a control diet; 0.0 HS = high ambient temperature + a control diet; PO = pequi oil.; 1.5 HS = high ambient temperature + a control diet supplemented with 1.5 g/kg PO; 3.0 HS = high ambient temperature + a control diet supplemented with 3.0 g/kg PO; 4.5 HS = high ambient temperature + a control diet supplemented with 4.5 g/kg PO; and 6.0 HS = high ambient temperature + a control diet supplemented with 6.0 g/kg PO.

**Table 1 animals-13-01896-t001:** Fatty acid and carotenoid composition of pequi oil (*C. brasiliense* Camb.).

Fatty Acid (%) ^a^						Carotenoids (µg/dL) ^b^		
Satured		Monounsaturated		Polyunsaturated		β-carotene	Lutein	Lycopene
Palmitic	37.03	Oleic (ω 9)	52.98	Linoleic (ω 6)	1.42	167.16	19.57	3.76
Lauric	2.28	Palmitoleic	0.17	Linolenic (ω 3)	1.01			
Stearic	0.31							
Araquidic	0.23							

^a^ Method described by Hartman and Lago [17]. ^b^ Method described by Correa et al. [18].

**Table 2 animals-13-01896-t002:** Ingredient and chemical composition of the experimental diets (pre-starter diets, provided during days 1–7; starter diets, provided during days 8–21).

Ingredients (%)	Levels of Pequi Oil									
	Pre-Starter					Starter				
	CON	1.5	3.0	4.5	6.0	CON	1.5	3.0	4.5	6.0
Soybean meal (CP 46%)	42.235	42.244	42.253	42.261	42.270	39.249	39.258	39.266	39.275	39.284
Corn	51.792	51.743	51.693	51.644	51.594	54.994	54.945	54.895	54.845	54.796
Soybean oil	1.776	1.667	1.558	1.449	1.340	1.929	1.820	1.711	1.602	1.494
Pequi oil	-	0.150	0.300	0.450	0.600	-	0.150	0.300	0.450	0.600
DL-Methionine (99%)	0.336	0.336	0.336	0.336	0.336	0.315	0.315	0.315	0.315	0.315
L-Lysine-HCL (99%)	0.125	0.125	0.125	0.125	0.125	0.139	0.139	0.139	0.139	0.139
L-Threonine (98.5%)	0.058	0.058	0.058	0.058	0.058	0.056	0.056	0.056	0.056	0.056
Dicalcium phosphate	1.872	1.872	1.872	1.872	1.872	1.604	1.604	1.604	1.604	1.604
Limestone	0.886	0.886	0.886	0.886	0.886	0.826	0.826	0.826	0.826	0.826
Coccidiostat (Coxistac 12%) ^3^	0.055	0.055	0.055	0.055	0.055	0.055	0.055	0.055	0.055	0.055
Choline chloride (60%)	0.100	0.100	0.100	0.100	0.100	0.100	0.100	0.100	0.100	0.100
Vitamin premix ^1^	0.100	0.100	0.100	0.100	0.100	0.100	0.100	0.100	0.100	0.100
Mineral premix ^2^	0.100	0.100	0.100	0.100	0.100	0.100	0.100	0.100	0.100	0.100
Sodium bicarbonate	0.114	0.114	0.114	0.114	0.114	0.083	0.083	0.083	0.083	0.083
Salt	0.450	0.450	0.450	0.450	0.450	0.450	0.450	0.450	0.450	0.450
Total	100	100	100	100	100	100	100	100	100	100
Calculation of nutrients										
Metabolizable energy, kcal/kg	2950	2950	2950	2950	2950	3000	3000	3000	3000	3000
Crude Protein, %	24.07	24.07	24.07	24.07	24.07	22.93	22.93	22.93	22.93	22.93
Calcium, %	0.96	0.96	0.96	0.96	0.96	0.86	0.86	0.86	0.86	0.86
Phosphorus (Available), %	0.47	0.47	0.47	0.47	0.47	0.41	0.41	0.41	0.41	0.41
Potassium, %	0.94	0.94	0.94	0.94	0.94	0.89	0.89	0.89	0.89	0.89
Sodium, %	0.22	0.22	0.22	0.22	0.22	0.21	0.21	0.21	0.21	0.21
Chlorine, %	0.34	0.34	0.34	0.34	0.34	0.34	0.34	0.34	0.34	0.34
Lysine (digestible), %	1.43	1.43	1.43	1.43	1.43	1.36	1.36	1.36	1.36	1.36
Methionine (digestible), %	0.68	0.68	0.68	0.68	0.68	0.64	0.64	0.64	0.64	0.64
Methionine + Cystine (digestible), %	1.06	1.06	1.06	1.06	1.06	1.01	1.01	1.01	1.01	1.01
Threonine (digestible), %	0.99	0.99	0.99	0.99	0.99	0.94	0.94	0.94	0.94	0.94

^1^ Provided per kg of diet: Vitamin A (min) 10,000 U.I.; Vitamin D3 (min) 2500 U.I.; Vitamin E (min) 17.5 U.I.; Vitamin K3 (min) 2 mg; Thiamin (min) 1.5 mg; Riboflavin (min) 7.5 mg; Niacina (min) 25 mg; Pantothenic acid (min) 15 mg; Pyridoxine, (min) 2 mg; Folic acid (min) 0.5 mg; Biotin (mín) 60 mcg; and B12 (min) 15 mcg. ^2^ Provided per kg of diet: Iron (min) 30 mg; Copper (min) 6 mg; Manganese (min) 70 mg; Zinc (min) 50 mg; Iodine (min) 1 mg; Selenium (min) 0.35 mg; and Cobalt (min) 0.2 mg. ^3^ Salinomycin (66 ppm).

**Table 3 animals-13-01896-t003:** Ingredient and chemical compositions of the experimental diets (grower diets, provided during days 22–35; finisher diets, provided during days 36–42).

Ingredients (%)	Levels of Pequi Oil									
	Grower					Finisher				
	CON	1.5	3.0	4.5	6.0	CON	1.5	3.0	4.5	6.0
Soybean meal (CP 46%)	32.239	32.248	32.256	32.265	32.274	26.969	26.977	26.986	26.995	27.003
Corn	61.977	61.927	61.878	61.828	61.778	68.121	68.071	68.021	67.972	67.922
Soybean oil	2.273	2.164	2.055	1.946	1.837	1.920	1.812	1.703	1.594	1.485
Pequi oil	-	0.150	0.300	0.450	0.600	-	0.150	0.300	0.450	0.600
DL-Methionine (99%)	0.279	0.279	0.279	0.279	0.279	0.239	0.239	0.239	0.239	0.239
L-Lysine-HCL (99%)	0.194	0.194	0.194	0.194	0.194	0.218	0.218	0.217	0.217	0.217
L-Threonine (98.5%)	0.064	0.064	0.064	0.064	0.064	0.058	0.058	0.058	0.058	0.058
Dicalcium phosphate	1.423	1.423	1.423	1.423	1.423	1.048	1.048	1.048	1.048	1.048
Limestone	0.697	0.697	0.697	0.697	0.697	0.667	0.667	0.667	0.667	0.666
Coccidiostat (Coxistac 12%) ^3^	0.055	0.055	0.055	0.055	0.055	-	-	-	-	-
Choline chloride (60%)	0.100	0.100	0.100	0.100	0.100	0.100	0.100	0.100	0.100	0.100
Vitamin premix ^1^	0.100	0.100	0.100	0.100	0.100	0.100	0.100	0.100	0.100	0.100
Mineral premix ^2^	0.100	0.100	0.100	0.100	0.100	0.100	0.100	0.100	0.100	0.100
Sodium bicarbonate	0.050	0.050	0.050	0.050	0.050	0.011	0.011	0.011	0.011	0.011
Salt	0.450	0.450	0.450	0.450	0.450	0.450	0.450	0.450	0.450	0.450
Total	100	100	100	100	100	100	100	100	100	100
Calculation of nutrients										
Metabolizable energy, kcal/kg	3100	3100	3100	3100	3100	3150	3150	3150	3150	3150
Crude Protein, %	20.25	20.25	20.25	20.25	20.25	18.28	18.28	18.28	18.28	18.28
Calcium, %	0.75	0.75	0.75	0.75	0.75	0.62	0.62	0.624	0.62	0.62
Phosphorus (Available), %	0.37	0.37	0.37	0.37	0.37	0.29	0.29	0.291	0.29	0.29
Potassium, %	0.79	0.79	0.79	0.79	0.79	0.71	0.71	0.712	0.71	0.71
Sodium, %	0.20	0.20	0.20	0.20	0.20	0.19	0.19	0.194	0.19	0.19
Chlorine, %	0.34	0.34	0.34	0.34	0.34	0.34	0.34	0.343	0.34	0.34
Lysine (digestible), %	1.22	1.22	1.22	1.22	1.22	1.10	1.10	1.101	1.10	1.10
Methionine (digestible), %	0.57	0.57	0.57	0.57	0.57	0.51	0.51	0.512	0.51	0.51
Methionine + Cystine (digestible), %	0.90	0.90	0.90	0.90	0.90	0.81	0.81	0.814	0.81	0.81
Threonine (digestible), %	0.84	0.84	0.84	0.84	0.84	0.76	0.76	0.759	0.76	0.76

^1^ Provided per kg of diet: Vitamin A (min) 10,000 U.I.; Vitamin D3 (min) 2500 U.I.; Vitamin E (min) 17.5 U.I.; Vitamin K3 (min) 2 mg; Thiamin (min) 1.5 mg; Riboflavin (min) 7.5 mg; Niacina (min) 25 mg; Pantothenic acid (min) 15 mg; Pyridoxine, (min) 2 mg; Folic acid (min) 0.5 mg; Biotin (min) 60 mcg; and B12 (min) 15 mcg. ^2^ Provided per kg of diet: Iron (min) 30 mg; Copper (min) 6 mg; Manganese (min) 70 mg; Zinc (min) 50 mg; Iodine (mín) 1 mg; Selenium (min) 0.35 mg; and Cobalt (min) 0.2 mg. ^3^ Salinomycin (66 ppm).

**Table 4 animals-13-01896-t004:** Averages and standard deviations for the temperature (°Celsius) and relative humidity (%) in the thermoneutral (TN) and heat stress (HS) ambient temperature environments.

Average ± Standard Deviation		
Items	Temperature	Relative Humidity
1 to 7 D		
TN Room	30.97 ± 0.63	64.45 ± 4.42
HS Room	34.51 ± 1.10	53.57 ± 4.09
8 to 21 D		
TN Room	27.30 ± 0.93	71.05 ± 3.90
HS Room	31.73 ± 0.97	59.57 ± 3.38
22 to 35D		
TN Room	24.06 ± 0.86	73.23 ± 3.79
HS Room	30.76 ± 1.19	62.14 ± 2.69
36 to 42 D		
TN Room	24.60 ± 1.27	77.20 ± 2.14
HS Room	29.07 ± 1.68	68.88 ± 1.65

**Table 5 animals-13-01896-t005:** Primer sequences used for real-time PCR ^1^.

Gene	Primers Sequences (5′-3′)	Gene Bank Number	Product Size (pb)
Hsp 70	F:CGGGCAAGTTTGACCTAA	NM_001006685.1	145
	R:TTGGCTCCCACCCTATCTCT		
Nrf2	F:GGAAGAAGGTGCTTTTCGGAGC	NM_205117.1	171
	R:GGGCAAGGCAGATCTCTTCCAA		
β-actina	F: TGCTGTGTTCCCATCTATCG	NM_205518.1	150
	R:TTGGTGACAATACCGTGTTCA		

^1^ HSP70 = Heat shock protein 70; Nrf2 = nuclear factor erythroid 2-related factor. F: Forward, R: Reverse.

**Table 6 animals-13-01896-t006:** Effects of temperature and dietary Pequi oil levels (PO level) on the serum biochemical indices in broilers exposed to heat stress.

Temperature ^1^	PO ^2^ Level (g/kg)		21 d						42 d				
		Glucose (mg/dL)	CHO (mg/dL)	TG (mg/dL)	HDL (mg/dL)	AST (U/L)	ALT (U/L)	Glucose (mg/dL)	CHO (mg/dL)	TG (mg/dL)	HDL (mg/dL)	AST (U/L)	ALT (U/L)
TN		207.22 ^b^	121.050	43.625 ^b^	118.908 ^b^	215.011	4.845 ^b^	226.032	124.039 ^a^	149.193	126.726	327.785 ^b^	3.478 ^b^
HS		219.91 ^a^	121.550	68.325 ^a^	140.578 ^a^	218.032	5.936 ^a^	220.037	103.371 ^b^	162.911	116.710	382.578 ^a^	4.764 ^a^
SEM		3.07	2.48	2.71	13.34	6.88	0.28	5.21	5.14	9.66	8.68	13.16	0.34
	CON	212.481	117.812 ^ab^	53.937	144.314	239.390	4.910	217.056	112.562	149.045	98.722	364.175	4.656
	1.5	211.250	112.687 ^b^	54.562	163.760	223.168	5.456	212.237	126.660	166.821	133.051	369.800	4.689
	3.0	214.210	130.500 ^a^	51.375	128.793	202.762	5.128	230.369	101.562	148.169	128.422	344.400	4.427
	4.5	216.137	117.250 ^ab^	56.062	132.471	205.375	5.675	220.013	121.187	139.937	123.406	350.466	3.741
	6.0	213.744	128.250 ^a^	63.937	104.380	211.911	5.784	235.500	106.553	176.287	123.836	347.066	3.093
SEM		4.63	3.37	4.71	19.72	9.22	0.41	8.01	7.96	15.15	13.67	20.73	0.55
TN	CON	202.550	118.250	40.500	139.911	263.143	4.146	226.300	131.625	131.714	119.028	355.300	4.074
	1.5	211.062	113.750	43.625	128.606	216.512	5.893	194.537	128.571	156.500	108.992	335.940	3.143
	3.0	202.112	127.000	43.750	119.518	180.500	3.928	236.720	107.250	143.625	152.553	315.150	4.490
	4.5	210.262	117.750	43.375	112.068	196.863	5.020	230.150	140.500	143.750	116.955	327.000	3.243
	6.0	210.112	128.500	46.875	94.440	218.037	5.238	242.450	112.250	170.375	133.796	305.533	2.444
HS	CON	222.412	117.375	67.375	148.717	215.637	5.675	207.813	93.500	166.375	78.416	373.050	5.238
	1.5	211.437	111.625	65.500	148.915	229.825	5.020	229.937	124.750	177.143	157.109	403.660	6.237
	3.0	226.312	134.000	59.000	138.036	225.025	6.329	224.012	95.875	152.714	104.291	373.650	4.365
	4.5	222.012	116.750	68.750	152.873	213.887	6.330	209.875	101.875	136.125	129.857	373.933	4.240
	6.0	217.375	128.000	81.000	114.321	205.786	6.329	228.550	100.857	182.200	113.877	388.600	3.741
SEM		6.77	5.03	5.89	30.43	13.12	0.62	10.76	10.46	21.75	18.53	30.57	1.86
*p*-value (factorial design)													
Temperature		0.006	0.880	<0.001	0.042	0.486	0.024	0.400	0.005	0.337	0.432	0.012	0.015
PO levels		0.969	0.003	0.392	0.436	0.141	0.725	0.218	0.178	0.491	0.495	0.913	0.230
Interaction effect		0.461	0.909	0.701	0.874	0.073	0.247	0.081	0.388	0.902	0.087	0.963	0.348
Effect of PO level ^3^													
Linear		0.655	0.277	0.084	0.115	0.048	0.173	0.087	0.538	0.659	0.484	0.685	0.020
Quadratic		0.864	0.930	0.752	0.496	0.104	0.354	0.692	0.709	0.418	0.253	0.513	0.537

^1^ TN = Thermoneutral; HS = Broiler chickens raised under cyclical heat stress for 8 h/day from 1 to 42 days of age. ^2^ CON (basal diet without PO); 1.5 (basal diet containing 1.5 g PO/kg diet); 3.0 (basal diet containing 3.0 g PO/kg diet); 4.5 (basal diet containing 4.5 g PO/kg diet); and 6.0 (basal diet containing 6.0 g PO/kg diet). ^a, b^ Means (n = 8) within each column with no common superscript differ (*p* < 0.05). CHO = total cholesterol; TG = triglycerides; and high-density lipoprotein = HDL. AST = aspartate aminotransferase; ALT = alanine aminotransferase; and SEM = standard error of the mean. ^3^ Orthogonal polynomials were used to evaluate linear and quadratic responses to the PO inclusion level.

**Table 7 animals-13-01896-t007:** Effects of temperature and dietary pequi oil (PO) levels on blood leukocyte profiles in broilers exposed to heat stress at 42 days of age.

Temperature ^1^	PO ^2^ Level (g/kg)	Heterophil (%)	Lymphocyte (%)	Eosinophils (%)	Basophils (%)	Monocytes (%)	H/L
TN		32.200	52.500	6.675	3.875	4.775	0.638 ^b^
HS		34.750	50.225	5.675	4.425	4.925	0.758 ^a^
SEM		1.14	1.46	0.78	0.43	0.73	0.042
	CON	31.813	52.25	7.625	2.75 ^b^	5.562	0.654
	1.5	36.500	46.812	6.312	5.25 ^a^	5.125	0.833
	3.0	31.187	52.625	5.437	5.312 ^a^	5.500	0.616
	4.5	34.937	51.437	6.062	3.187 ^ab^	4.375	0.746
	6.0	32.927	53.687	5.437	4.25 ^ab^	3.687	0.641
SEM		1.78	2.29	1.19	0.64	1.14	0.07
TN	CON	28.500	54.875	6.375	3.000	7.250	0.539
	1.5	32.000	50.375	9.125	5.125	3.375	0.654
	3.0	32.000	51.375	5.500	4.750	6.500	0.662
	4.5	35.750	52.625	4.750	2.625	4.250	0.693
	6.0	32.750	53.250	7.625	3.875	2.500	0.641
HS	CON	35.125	49.625	8.875	2.500	3.875	0.768
	1.5	41.000	43.250	3.500	5.375	6.875	1.012
	3.0	30.375	53.875	5.375	5.875	4.500	0.570
	4.5	34.125	50.250	7.375	3.750	4.500	0.799
	6.0	33.125	54.125	3.250	4.625	4.875	0.641
SEM		2.37	3.14	1.46	0.91	1.52	0.09
*p*-value (factorial design)							
Temperature		0.105	0.280	0.360	0.359	0.886	0.046
PO levels		0.177	0.277	0.702	0.022	0.755	0.127
Interaction effect		0.099	0.565	0.050	0.898	0.204	0.136
Effect of PO level ^3^							
Linear		0.904	0.311	0.250	0.651	0.230	0.606
Quadratic		0.524	0.339	0.544	0.415	0.651	0.392

^1^ TN = Thermoneutral; HS = Broiler chickens raised under cyclical heat stress for 8 h/day from 1 to 42 days of age. ^2^ CON (basal diet without PO); 1.5 (basal diet containing 1.5 g PO/kg diet); 3.0 (basal diet containing 3.0 g PO/kg diet); 4.5 (basal diet containing 4.5 g PO/kg diet); and 6.0 (basal diet containing 6.0 g PO/kg diet). Means (n = 8) within each column with no common superscript letter differ (*p* < 0.05). H/L = heterophil-to-lymphocyte ratio; and SEM = standard error of the mean. ^3^ Orthogonal polynomials were used to evaluate linear and quadratic responses to PO inclusion levels.

**Table 8 animals-13-01896-t008:** Effects of temperature and dietary pequi oil (PO) levels on relative organ weights (%) in broilers exposed to heat stress.

Temperature ^1^	PO Level ^2^ (g/kg)	21 Days					42 Days				
		Spleen	Bursa	Thymus	Pancreas	Liver	Spleen	Bursa	Thymus	Pancreas	Liver
TN		0.077	0.226	0.436	0.298	2.239 ^a^	0.108	0.136	0.303	0.151	1.777
HS		0.079	0.211	0.383	0.308	2.149 ^b^	0.103	0.130	0.274	0.154	1.804
SEM		0.003	0.009	0.020	0.007	0.030	0.004	0.007	0.018	0.004	0.035
	CON	0.078	0.206	0.379	0.303	2.204	0.101	0.130	0.265	0.155	1.784
	1.5	0.080	0.227	0.382	0.294	2.132	0.108	0.147	0.284	0.147	1.722
	3.0	0.077	0.222	0.458	0.307	2.175	0.108	0.120	0.290	0.145	1.714
	4.5	0.079	0.217	0.400	0.300	2.197	0.104	0.135	0.296	0.153	1.879
	6.0	0.077	0.220	0.428	0.311	2.263	0.105	0.134	0.309	0.162	1.856
SEM		0.004	0.013	0.031	0.010	0.048	0.007	0.011	0.029	0.053	0.007
TN	CON	0.077	0.252	0.388	0.307	2.361	0.114	0.142	0.316	0.165	1.844
	1.5	0.081	0.215	0.393	0.280	2.163	0.111	0.139	0.300	0.144	1.678
	3.3	0.072	0.234	0.523	0.304	2.190	0.110	0.125	0.346	0.151	1.683
	4.5	0.076	0.213	0.407	0.303	2.222	0.110	0.128	0.275	0.137	1.856
	6.0	0.081	0.218	0.466	0.299	2.258	0.093	0.148	0.278	0.160	1.827
HS	CON	0.079	0.159	0.369	0.300	2.047	0.088	0.117	0.213	0.146	1.724
	1.5	0.079	0.239	0.370	0.307	2.101	0.105	0.155	0.268	0.150	1.766
	3.3	0.082	0.211	0.393	0.311	2.160	0.106	0.116	0.234	0.138	1.744
	4.5	0.083	0.222	0.393	0.298	2.171	0.099	0.142	0.315	0.170	1.902
	6.0	0.073	0.222	0.390	0.324	2.268	0.012	0.119	0.339	0.164	1.884
SEM		0.065	0.019	0.042	0.015	0.063	0.009	0.016	0.039	0.009	0.075
*p*-value (factorial design)											
Temperature		0.653	0.198	0.065	0.356	0.032	0.449	0.544	0.265	0.694	0.594
PO levels		0.976	0.841	0.337	0.847	0.379	0.943	0.625	0.864	0.379	0.128
Interaction effect		0.633	0.251	0.634	0.752	0.108	0.113	0.566	0.120	0.058	0.687
Effect of PO level ^3^											
Linear		0.822	0.680	0.243	0.521	0.235	0.826	0.924	0.285	0.364	0.084
Quadratic		0.793	0.481	0.481	0.626	0.164	0.534	0.934	0.911	0.086	0.223

^1^ TN = Thermoneutral; HS = Broiler chickens raised under cyclical heat stress for 8 h/day from 1 to 42 days of age. ^2^ CON (basal diet without PO); 1.5 (basal diet containing 1.5 g PO/kg diet); 3.0 (basal diet containing 3.0 g PO/kg diet); 4.5 (basal diet containing 4.5 g PO/kg diet); and 6.0 (basal diet containing 6.0 g PO/kg diet). ^a,b^ Means (n = 8) within each column with no common superscript letter differ (*p* < 0.05). SEM = standard error of the mean. ^3^ Orthogonal polynomials were used to evaluate linear and quadratic responses to PO inclusion levels.

**Table 9 animals-13-01896-t009:** Effects of temperature and dietary pequi oil (PO) levels on antioxidant enzyme activity in broilers exposed to heat stress at 42 days of age.

Temperature ^1^	PO Level ^2^ (g/kg)	SOD (U/mg Protein)	CAT (U/mg Protein)	GPx (U/mg Protein)
TN		31.04 ^b^	1.494	8.199 ^a^
HS		33.61 ^a^	1.672	5.553 ^b^
SEM		0.730	0.130	0.190
	CON	32.870	1.670	7.090
	1.5	31.830	1.731	6.697
	3.0	32.235	1.663	6.500
	4.5	32.010	1.486	7.522
	6.0	32.677	1.378	6.660
SEM		1.23	0.50	0.99
TN	CON	32.489	1.366	8.167
	1.5	29.850	1.755	8.187
	3.0	29.651	1.495	6.990
	4.5	31.938	1.738	9.645
	6.0	31.252	1.115	8.004
HS	CON	33.251	1.948	6.015
	1.5	33.805	1.706	5.028
	3.0	34.819	1.831	6.006
	4.5	32.080	1.234	5.399
	6.0	34.102	1.640	5.319
SEM		1.64	0.31	0.78
*p*-value (factorial design)				
Temperature		0.010	0.363	<0.001
PO Diet		0.956	0.790	0.572
Interaction effect		0.438	0.392	0.211
Effect of PO level ^3^				
Linear		0.521	0.309	0.974
Quadratic		0.848	0.587	0.873

^1^ TN = Thermoneutral; HS = Broiler chickens raised under cyclical heat stress for 8 h/day from 1 to 42 days of age; ^2^ CON (basal diet without PO); 1.5 (basal diet containing 1.5 g PO/kg diet); 3.0 (basal diet containing 3.0 g PO/kg diet); 4.5 (basal diet containing 4.5 g PO/kg diet); and 6.0 (basal diet containing 6.0 g PO/kg diet). ^a,b^ Means (n = 8) within each column with no common superscript letter differ (*p* < 0.05). SOD = superoxide dismutase; CAT = catalase; GPx = glutathione peroxidase; and SEM = standard error of the mean. ^3^ Orthogonal polynomials were used to evaluate linear and quadratic responses to PO inclusion levels.

**Table 10 animals-13-01896-t010:** Effects of temperature and dietary pequi oil (PO) levels on serum MDA concentrations in broilers exposed to heat stress at 42 days of age.

Temperature ^1^	PO Level ^2^ (g/kg)	Serum MDA (nmol/L)
TN		8.950 ^b^
HS		13.42 ^a^
SEM		0.71
	CON	13.686
	1.5	10.718
	3.0	10.655
	4.5	10.294
	6.0	10.583
SEM		1.16
TN	CON	8.382 ^b^
	1.5	9.328 ^b^
	3.0	8.907 ^b^
	4.5	7.899 ^b^
	6.0	10.235 ^b^
HS	CON	18.991 ^a^
	1.5	12.108 ^ab^
	3.0	12.402 ^ab^
	4.5	12.689 ^ab^
	6.0	10.931 ^b^
SEM		1.30
*p*-value (factorial design)		
Temperature		<0.001
PO Diet		0.091
Interaction effect		0.012
Effect of PO level ^3^		
Linear TN		0.649
Quadratic TN		0.707
Linear HS		0.001
Quadratic HS		0.051

^1^ TN = Thermoneutral; HS = Broiler chickens raised under cyclical heat stress for 8 h/day from 1 to 42 days of age.^2^ CON (basal diet without PO); 1.5 (basal diet containing 1.5 g PO/kg diet); 3.0 (basal diet containing 3.0 g PO/kg diet); 4.5 (basal diet containing 4.5 g PO/kg diet); and 6.0 (basal diet containing 6.0 g PO/kg diet). ^a,b^ Means (n = 8) within each column with no common superscript letter differ (*p* < 0.05). SEM = standard error of the mean.^3^ Orthogonal polynomials were used to evaluate linear and quadratic responses to PO inclusion levels.

## Data Availability

The data presented in this study are available upon request from the corresponding author.

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
