# Peer review of "Pequi Oil (Caryocar brasiliense Camb.) Attenuates the Adverse Effects of Cyclical Heat Stress and Modulates the Oxidative Stress-Related Genes in Broiler Chickens"

_animals, 2023, doi:10.3390/ani13121896_

Round 1

Reviewer 1 Report

The presented study analyzes the impact

PO on Cyclical Heat Stress and Oxidative Stress Related Genes in Broiler Chickens. Losses due to high temperature during broiler rearing affect not only the health aspect of the flock, but also the final quality of the meat. The search for natural, easily available additives to feed or water that can reduce the impact of heat stress, reduce the accompanying oxidative stress, is becoming an increasingly common topic of scientific work, which gives a real chance for the future launch of preparations that actually work. The paper submitted for review is a detailed analysis of the impact of the local PO product on the problem of heat stress in broiler chickens. The experiment was planned and carried out correctly, the methodology used is correct, the authors, in addition to classic analyses, used a multi-directional approach, reaching for blood tests, assessment of oxidative stress markers and molecular biology, which certainly distinguishes the study from others in a similar field. The groups and their size are correct, which enables the performance of correct statistical analyses. Results, also in tabular form, presented in a correct, legible way. Conclusions corresponding to a reliable analysis of previous observations of the phenomenon of heat stress and oxidative stress are correctly put forward. The only remark regarding the methodology is that no analysis of production parameters for individual groups was performed, no histopathological evaluation of the organs subjected to weight analysis was performed - such analyzes and the results obtained would be a perfect complement to the study, and additionally they do not require large financial outlays.

Below are some comments and questions about the study.

L-144 During the experimental period (from 1 to 42 days of age), all broiler chickens 144 in the thermoneutral room (TN) were reared under temperature conditions recommended 145 by the Cobb Broiler Management Guide, but adapted to the Brazilian context.

Could you explain the term „Brazilian context” and give the details about this adaptation? Not all readers know the Brazilian way of rearing the chicken. Details will be helpful to understand this context.

Is there any commercial product with PO dedicated to poultry?

What, in your opinion, based on your observation will be the future of PO use in poultry?

What about economic aspects of PO use in commercial poultry market?

Did you make any analysis related to production parameters like FI, BW, FCR, EBI/EPEF? Are there any studies related to this kind of observation and PO ?

Did you make any histopathological analysis from organ samples? In case of use of natural feed additives, studies showed that they may cause lipid droplets in hepatocytes which may affect the organ metabolism etc. I wonder if PO would have any potentially negative effects on organs, and such a study would complement the analysis of organ size and blood biochemical parameters

References -please use the format recommended by the journal.

Author Response

We appreciate the contribution to our manuscript, and we believe that the comments made are relevant and important for the publication (animals-2382613). Our comments and responses to observations made by Reviewer 1 follow below:

Reviewer 1:

L-144 During the experimental period (from 1 to 42 days of age), all broiler chickens 144 in the thermoneutral room (TN) were reared under temperature conditions recommended 145 by the Cobb Broiler Management Guide, but adapted to the Brazilian context.

Could you explain the term Brazilian context” and give the details about this adaptation? Not all readers know the Brazilian way of rearing the chicken. Details will be helpful to understand this context.

A: In this temperature stress experiment we used temperature ranges that are common in different Brazilian states, but considering Cobb Broiler Management Guide.

Is there any commercial product with PO dedicated to poultry? What, in your opinion, based on your observation will be the future of PO use in poultry? What about economic aspects of PO use in commercial poultry market?

A: Pequi oil (PO) is a product still obtained by hand, carried out by communities in some states of Brazil (Goias, Minas Gerais for example). Interest in this research arose from some promising results seen in work with mice by Prof. Dr. Cesar Koppe. The study of pequi oil in broiler chicken feed as a potential functional oil is unprecedented, and we had good results that encourage us to research higher levels of the product and proteomic evaluations can be an interesting tool in this process. We believe that pequi oil can be promising as a functional oil, more and more favorable research is emerging on this product in the research human health, and we believe that further research in the animal area is necessary, and ultimately it can economically help communities in the exploitation rationale for this product.

Did you make any analysis related to production parameters like FI, BW, FCR, EBI/EPEF? Are there any studies related to this kind of observation and PO? Did you make any histopathological analysis from organ samples? In case of use of natural feed additives, studies showed that they may cause lipid droplets in hepatocytes which may affect the organ metabolism etc. I wonder if PO would have any potentially negative effects on organs, and such a study would complement the analysis of organ size and blood biochemical parameters

A: The study of pequi oil comprises a large study, and the performance results, biochemical indices, organ weight, blood leukocyte profile, among other analyzes will be contemplated in another publication.

References - please use the format recommended by the journal.

A: Citations, references, tables, and footnotes have been carefully revised. In addition, all changes made to the text are highlighted in the submitted file (revised paper).

Reviewer 2 Report

The present study examines the effects of pequi oil (PO) on the alleviation of heat stress in broiler chickens. It falls within the scope of the journal. The experimental design is appropriate. However, there are some issues (listed below) that must be addressed by the authors in order to improve the paper.

MAIN COMMENTS

1. Why didn't you measure antioxidant enzyme activity and MDA at 21 days of age? 

2. Why didn't you report the growth performance of the broilers? Was this a follow-up study? Reduced feed intake and poor weight gain are also very good indicators of the impact of heat stress in broilers. 

SPECIFIC COMMENTS

Lines 125-126: Why cages and not litter floor? Is it a national regulation or there was a specific purpose for this? Please check your data regarding the density of the animals again. A 0.5x0.6 cage does not give 4m2 per broiler (unless I missed something here)

Line 142: Natural matter? Do you mean as-fed basis? If yes, please correct.

Line 245: Delete ''Orthogonal''.

Line 342: Figure 1 is missing.

Lines 355-356: Where are these data in the paper? 

There are too many tables in the manuscript.

Table 1: I believe that the fatty acid composition of PO is irrelevant to the objective of the study. Only carotenoid content is. I would suggest to delete Table 1 and report the carotenoid content only in the manuscript.

Tables 2 and 3: They should be merged. Why did you use a 4-phase feeding?

Table 4: A temperature-humidity index (THI) would be a better measurement than temperature and humidity alone, for heat stress (or thermal comfort to be more precise) in broilers. Please use literature data to calculate THI.

Table 6: Please correct HE in the first column. It is HS.

Tables 9 and 10: They should be merged as data in both tables are relevant to the oxidative status of broilers.

English is generally good. Only a few parts in the discussion section must be revised.

Author Response

We appreciate the contribution to our manuscript, we believe that the comments made are relevant and important for the publication (animals-2382613). Our comments and responses to observations made by Reviewer 2 follow below.

Reviewer 2:

  1. Why didn't you measure antioxidant enzyme activity and MDA at 21 days of age?

A: This analysis followed the other analysis of enzymes of the antioxidant system in order to maintain a standard analysis and experimental logistics.

  1. Why didn't you report the growth performance of the broilers? Was this a follow-up study? Reduced feed intake and poor weight gain are also very good indicators of the impact of heat stress in broilers.

A: The study of pequi oil comprises a large study, and the performance results, biochemical indices, organ weight, blood leukocyte profile, among other analyzes will be contemplated in another recent publication.

Lines 125-126: Why cages and not litter floor? Is it a national regulation or there was a specific purpose for this? Please check your data regarding the density of the animals again. A 0.5 x 0.6 cage does not give 4m2 per broiler (unless I missed something here).

A: Done as requested

Line 142: Natural matter? Do you mean as-fed basis? If yes, please correct.

A: Completed

Line 245: Delete ''Orthogonal''.

A: Done as requested

Line 342: Figure 1 is missing.

A: Completed

Lines 355-356: Where are these data in the paper? 

A: Completed

There are too many tables in the manuscript.

Table 1: I believe that the fatty acid composition of PO is irrelevant to the objective of the study. Only carotenoid content is. I would suggest to delete Table 1 and report the carotenoid content only in the manuscript.

A: This study comprises several analyzes and with that some tables that best present the observed data. The study of pequi oil (PO) in broiler chicken feed is unprecedented, and we believe that the information in the table on PO can help in future comparisons in research in poultry and in target species of laboratory studies, we believe that the number citations of our work can be expanded in this way.

Tables 2 and 3: They should be merged. Why did you use a 4-phase feeding?

A: In this study, we applied 4 experimental phases to better meet nutritional requirements. Pequi oil was not added for energy purposes, but because it is an oil it has these properties (which were previously evaluated by our study group), and each diet and table presents a variation in the amount of soy oil, for this reason we present the 4 tables of the diets.

Table 4: A temperature-humidity index (THI) would be a better measurement than temperature and humidity alone, for heat stress (or thermal comfort to be more precise) in broilers. Please use literature data to calculate THI.

A: Unfortunately, we do not provide the calculation, as we do not have the dry bulb and wet bulb data to follow the Tao and Xin, 2003 formula for THI in broilers. But the temperature and humidity information presented follows publication standards based on animals subjected to heat stress.

Table 6: Please correct HE in the first column. It is HS.

A: Done as requested

Tables 9 and 10: They should be merged as data in both tables are relevant to the oxidative status of broilers.

A: The tables present different statistical results, in table 10 there was interaction, and we show the complete data, unfortunately it is not feasible to join the tables.

English is generally good. Only a few parts in the discussion section must be revised.

A: This manuscript has been proofread by native English speakers.